# Lived experiences of caregivers of persons with epilepsy attending an epilepsy clinic at a tertiary hospital, eastern Uganda: A phenomenological approach

**Lindah Okiah[1], Samuel Olowo[1¤]\*, Stanely J. Iramiot[2], Rebecca Nekaka[3], Lydia V. N. Ssenyonga[1]**

1 Department of Nursing, Busitema University Faculty of Health Sciences, Mbale, Uganda, 2 Department of Microbiology and Immunology, Busitema University Faculty of Health Sciences, Mbale, Uganda, 3 Department of Community and Public Health, Busitema University Faculty of Health Sciences, Mbale, Uganda

¤ Current address: Department of Nursing, Mbarara University of Science and Technology, Mbarara, Uganda

\* samuel28olowo@gmail.com

**Data Availability Statement:** All relevant data are within the manuscript and its Supporting

## Abstract

### Introduction

Epilepsy has been found to affect caregivers' quality of life, lifestyle, psychological health, social well-being, and working time. Caregivers in Uganda as in the rest of the world are important in assisting a person with epilepsy in complying with medical directions and are actively involved in communicating with healthcare professionals. Little is known about the lived experiences of caregivers of persons afflicted with epilepsy in Uganda. The purpose of this study was to determine the lived experiences of caregivers of persons with epilepsy attending the epilepsy clinic at Mbale regional referral hospital, eastern Uganda.

### Methods and materials

The caregivers' lived experiences were elicited directly from them and their health workers who work with them in the care of the patients. Forty participants which consisted of 30 care-givers and 10 key informant health workers were selected for the study through purposive sampling. Face-to-face in-depth interviews with an unstructured interview guide were conducted to gather participants' information. The principal investigator conceptualized the interview guide, the guide was then reviewed by co-investigators, and revised and approved as the final data collection instrument after an extensive and comprehensive literature review. The interview guide comprised two sections; the first section comprised the questions that elicited the participants' social-demographic information. The second section comprised questions that explored caregivers' experiences of persons afflicted with epilepsy. Notations were taken and a digital recorder was used purposely for audio recordings. All interviews lasted for an hour and were audio-recorded with the participant's consent. An

Information files. The supporting information files was submitted in the earlier revisions (first and second)

**Funding:** The author(s) received no specific funding for this work.

**Competing interests:** The authors have declared that no competing interests exist.

inductive thematic analysis was employed and adopted to identify the patterns emerging from the texts.

## Results

The caregivers majorly perceived epilepsy as a burden. Four main themes were revealed from the analysis and these are: psychological burdens which included, worries about the future of the patient, being looked down upon; social burdens which entailed, affected public relations, feelings of stigma; an economic burden which included interference with the source of income, affected productivity at work; and physical burdens which included, Feelings of uneasiness and disrupted sleep among others.

## Conclusion

The caregivers majorly perceived epilepsy as a serious burden. This burden can be psychological, social, economic, and physical. Therefore, services and plans targeting patients with epilepsy need to consider the burden that caregivers encounter to comprehensively manage epilepsy.

## Introduction

Epilepsy remains a global health challenge affecting all key healthcare players including caretakers [1]. The burden of refractory epilepsy is on the increase despite major pharmacologic advances in the management of epilepsy with over 22.5% of epileptic patients suffering from refractory epilepsy [1]. Studies have shown epilepsy to be common across the population, with devastating effects and increasing prevalence rates [2]. The World Health Organization (WHO) recognizes epilepsy as a chronic non-communicable disease and a significant public health challenge affecting over 50 million people worldwide [3]. Epilepsy accounted for 0.56% of the global burden of disease and therefore poses a major burden to the global healthcare community [4]. The disease burden from epilepsy has been reported to be higher in developing regions of the world [5]. According to the 2013 epidemiological report on sub-Saharan Africa, epilepsy was the fourth most common neurological condition among regions of Africa, with prevalence rates estimated to vary from 49 to 215 per 100,000 people [6]. The incidence of epilepsy has been reported to be high in Uganda with over 156 new cases per 100,000 person-years [7] and an increasing prevalence of 13.3% [8]. Epilepsy affects everyone irrespective of race, culture, ethnicity, country, time, and space [4]. Epilepsy and its resultant burden directly affect relationships between the caregiver and the care receiver [9]. It has a significant implication for the individual affecting the quality of life and indirectly affecting the caregivers in various life domains [1]. The burden of epilepsy does not only arise from seizure activities but other factors like social and cultural stigma affiliated with epilepsy [10]. The social rejection of epilepsy victims, and being barred from marriage, and employment have been associated with an increased burden of epilepsy in the community [11]. In East and South-central Africa, stigma from epilepsy has been associated with low school attendance, low school attainment among children, and increased inferiority complex among caregivers in society [1]. Caregiving and receiving care for epilepsy can happen at any moment in life. "It involves a multidimensional role, varying from daily routine activities such as household management, checking and monitoring tasks, providing transportation and physical care to recognizing

reportable symptoms" [12]. Caregivers can further assist persons with epilepsy in complying with medical directions and can be actively involved in communicating with healthcare professionals [13, 14]. The caregivers of persons with epilepsy experience severe emotional, physical, and economic burdens resulting from the nature, chronicity, disability, and stigma affiliated with epilepsy condition [15–19]. Several studies globally have shown that a substantial burden is impacted in various ways on caregivers who support individuals with epilepsy. While epilepsy is a common phenomenon in Uganda, there has been limited regional research done to study the lived experiences of caregivers of persons with epilepsy. Moreover, concerning the Health Management and Information Systems (HMIS) register at Mbale regional referral hospital psychiatric unit, there was a paucity of information concerning lived experiences of caregivers of persons with epilepsy. Therefore, this study aimed to determine the lived experiences of caregivers of persons with epilepsy attending the epilepsy clinic at Mbale regional referral hospital, eastern Uganda.

## Materials and methods

### Study design

This study employed a qualitative phenomenological approach. A Phenomenology is an approach to qualitative inquiry that focuses on lived experiences of persons [20]. Through using this approach, the researcher gains a profound understanding of the phenomenon. Therefore, it generates a detailed view of the human experience.

### Study site

The study was conducted at Mbale regional referral hospital Psychiatric Unit. Mbale regional referral hospital is located in Eastern Uganda approximately 214km East of Kampala, Uganda's capital [21]. Mbale regional referral hospital receives patients of all age groups children, men, and women who come from all over the 15 districts of the eastern region of the country [22]. It has a catchment population of 4 million, a bed capacity of 511, and 18 inpatient wards [22]. The hospital has over forty physicians including intern doctors, over 170 nurses, and students (both medical and nursing) [23]. The psychiatric unit has an inpatient and runs an outpatient epileptic clinic every Wednesday, patients attending this clinic usually come along with their caregivers. The psychiatric unit has no trained medical doctors working in the unit [24]. The psychiatric unit is majorly covered by psychiatric clinical officers who hold a diploma in clinical medicine and with a psychiatry specialization and psychiatric nurses with diplomas and certificates in psychiatry specialization [24].

### Study population

The caregivers' lived experiences were elicited directly from them and their health workers who work with them in the care of the patients. Therefore, the study population comprised 30 adult caregivers of patients with epilepsy who attended the epileptic clinic at Mbale regional referral hospital and 10 health workers who attended to patients with epilepsy at the hospital during the study period. This gives a total population of 40 participants for the study.

### Eligibility criteria

The caregivers who were eligible for the study were those who were at least 18 years and lived with the patient. Furthermore, the caregivers recruited were those involved in close monitoring of the patient in terms of treatment adherence and accompanied the patient regularly for

the follow-up clinics. This study excluded caregivers whose persons had other physical and mental disorders in addition to epilepsy.

For the key informants, the study enrolled health workers who had worked in the psychiatric unit for at least six months and were regularly involved in the clinical care of persons with epilepsy.

Finally, caregivers and Key informants who declined informed consent were excluded from the study.

## Sampling technique

Purposive sampling was employed to select only caretakers of persons living with epilepsy. For the sample size, saturation sampling was employed. Using this approach, saturation was assumed when no new information emerged. Full saturation was reached on a sample of 30 participants. This sample size was adequate as evidence has shown that code saturation can be reached with nine in-depth interviews and 16 to 24 in-depth interviews are needed to reach meaning saturation [25]. The code saturation meant that the "researcher has heard it all" while the meaning saturation meant that the researcher has "understood it all" [25]. Therefore, the sample size of 30 caregivers in this study was adequate to reach meaningful saturation. The health workers, who were the key informants were purposively selected. Saturation sampling was further employed on the key informants and saturation was reached with 10 in-depth interviews. However, stratification was ensured while interviewing the participants to include psychiatric nurses of which they were six, two psychiatric clinical officers, one counselor, and one peer support. The aim was to ensure an accurate and in-depth understanding of the lived experiences of the caregivers from the perspective of the different healthcare professionals involved in their care since the management of epilepsy requires an interprofessional approach.

## Data collection

Face-to-face in-depth interviews with an unstructured interview guide were conducted to gather participants' information. An unstructured interview [26] is a "conversation with a purpose" and enables researchers to obtain in-depth information. "It is also described as a shared experience in which researchers and interviewees come together to create a context of conversational intimacy in which participants feel comfortable telling their story". The interview guide consisted of open-ended questions and follow-up probes to ensure a better understanding. The aim was to ascertain and explore lived experiences about the care of a patient with epilepsy. As part of the study's quality control measures, the interview guide was pretested among four caregivers attending the same clinic who consented to take part in the pilot study. The rationale of the pilot was to check for the average length of the interview, language suitability, and potential sources of bias like leading questions. The pilot data was transcribed to test whether it captured enough relevant data to answer the study's aim.

Necessary adjustments were later done to ensure clarity. Results from the pilot survey were not considered during the final analysis. Before the interview began, verbal and written informed consent was obtained from the participants. An understanding was ensured among the prospective participant of the aims and scope of the study. The interview guide was conceptualized by the investigator, reviewed by co-investigators, and revised and approved as the final data collection instrument. This was after a thorough literature review. Distractions during the interview process were avoided as much as possible by conducting the interviews in quiet secluded rooms. The quiet rooms were assumed to be a neutral setting and would encourage freedom of speech among the participants. The presence of a trained and

experienced qualitative researcher was ensured during the interview. The experienced qualitative researcher also assisted in taking note of the non-verbal cues. The interview lasted between 40 minutes to 1hour. All caregivers were interviewed in the language they best understood which was English, Lugisu, or Luganda. The key informants (health workers) were interviewed in English. During interview guide preparations, a fruitful discussion was ensured for those English words that had no direct translation in the local languages, and suitable ways of conveying them to participants were reached. Saturation was reached when no new information was emerging among the participants. The interviewers pre-briefed, had frequent reviews and debriefed each other throughout the interview cycle. All that was ensured is the preservation of the original meaning of experiences as conveyed by the participants.

## Data analysis and presentation

The In-depth interviews were audio recorded and later subjected to a careful verbatim transcription.

This study employed Braun and Clarke's six-phase approach to thematic analysis [27] to analyze the data. The phases are "(i) familiarizing the researcher with the data, (ii) generating the initial codes, (iii) searching for themes, (iv) reviewing themes, (v) defining and naming themes, and (vi) producing a report".

Data were presented narratively. The reporting of data was per the consolidated criteria for reporting qualitative research (COREQ) guidelines [28].

## Rigor and trustworthiness of the study

The study employed trustworthiness based on Lincoln and Guba's criteria which entails criteria of credibility, dependability, conformability, and transferability [29]. Credibility refers to the confidence that can be portioned to the research finding. It involves ensuring that the results are believable, consistent with reality, and that the interpretations are true [29]. In this study, credibility was ensured through independent analysis of the data by the investigators and a trained and experienced qualitative researcher. On a later comparison of the analysis results, a high agreement was demonstrated.

Confirmability: Confirmability refers to a measure of how well the study's discoveries are supported by the data collected and reflects the objectivity of the data. This means that the researcher's bias should not alter the result [29]. Confirmability was ensured by recording of every activity of the participants during the time of the interview. In addition, the audio-taped interviews were not destroyed which can enable others to track the process. Moreover, data analysis, interpretations, and conclusions of findings were shared with experienced qualitative researchers for peer debriefing before synthesizing the final outputs. It was also achieved by using quotes which means linking the words of the participants with the discoveries [29]. This can be further demonstrated by how the study findings are in line with the findings of other related studies.

Dependability: Dependability entails the evaluation of the data collection and data analysis process [29]. In this study, dependability was ensured by ensuring clarity in the research process.

Transferability: Transferability refers to the degree to which the results of qualitative research can be transferred to other contexts or settings with other respondents [29]. Transferability was ensured through the thick description of the study in terms of the study setting, and data collection procedure, involving a trained and experience qualitative researcher to take note of the non-verbal cues and debriefing after every interview.

## Ethical considerations

Ethical approval and permission to conduct the study were sought from Cure Children's Hospital of Uganda Research and Ethics Committee (Approval number: CCHU-REC/08/019). Strict Adherence to the ethical principles for medical research involving human subjects as stipulated in the Declaration of Helsinki [30] and the ethical principles of the Belmont Report of respect for persons, beneficence, and justice [31] were ensured throughout the entire research process.

Participants voluntarily consented to take part in the study after a detailed explanation of the aim of the study. Participants were further informed that any decision to withdraw from the study will not have any impact on the healthcare services rendered to their patients or a negative impact on their profession for the case of the key informants.

## Results

### Demographics of the caregivers

Out of the thirty caregivers that were interviewed, 16(53%) were females and 14(47%) of the participants were males. Majority, 20(67%) of the participants were married, 9(30%) were single and 1(3%) of the participants was divorced. With regards to education, the majority, 15 (50%) of the participants never acquired formal education, 11(37%) acquired certificate level education, 3(10%) acquired a diploma level in education and 1(3%) had degree-level education. With regards to tribes, the majority, 20(67%) of the caregivers were Bagishu, and the remaining 10(33%) of the participants were of different tribes. Professionally, 29(97%) of the participants comprised security guards, farmers, students, businessmen, and women, and 1 (3%) of the participants was a nurse. About 10(33%) of the participants were biological fathers, 3(10%) of the participants were sisters to the patient, 13(43%) of the participants were biological mothers, 1(3%) was a spouse and 2(7%) of the participants were caregivers. About 13(43%) of the participants reported that their patients had epilepsy for more than 5 years, 8(27%) fell in the range between three to five years, 8(27%) of the participants fell in the range between one to two years and 10(33%) of the participants reported that their participants had suffered from epilepsy for less than a year.

### Burden of epilepsy

The caregivers perceived epilepsy as a major burden affecting different aspects of their lives: psychosocial burden social burden, economic and physical burdens summarized in Tables 1–3 respectively.

**Psychological burden.** This is a situation where a caregiver was assessed with a tendency to get a stable mind, which is perceived as a threatening or excessive uneasiness. The psychological burden was discussed under; the effects of caring for persons afflicted with epilepsy in day-to-day life, interference with a personal relationship with others, time for one's self, feelings after separation from the patients, and behavioral influence.

*Effects of caring for persons afflicted with epilepsy in day-to-day life.* The psychological burden was expressed differently among caregivers, with excessive uneasiness resulting from worries about potential daily attacks, concern about the future lives of their persons, worries about incurability of epilepsy, and thought blocks on what to do especially when the attack occurs even after taking medications, were conceptualized among the participants, illustrated in Table 1.

**Table 1. Psychological burden experienced by caregivers.**

| Major Themes | Sub-themes | Selected illuminative quotes |
|---|---|---|
| **Psychological Burden** | **Effects on day-to-day life** | *"……I worry a lot that this child will die any time, I think a lot because all the time she gets sick even right now we are hospitalized here but when we are discharged, I know she is going to get an attack at any time that makes me worry…………."* [Caregiver no.7]. |
| | | *"As for me when she falls, I feel bad, the last episode which happened recently she fell the whole day and stayed hungry the whole day she didn't eat. mmh………if she is fine she can dig or even fetch water………."* [Caregiver no.2]. |
| | **Effects on the personal relationship** | *"… Some people fear me and they isolate themselves from me, some of them fear my son and think that he will infect their children nowadays I restrict him from moving to the neighbors' homes, we stay in our home…"* [Caregiver no.4]. |
| | | *"…Mmmh…we talk well with my friends and neighbors, there are some days when he gets an attack when I am not around but my friends and neighbors who like me come around and help me…"* [Caregiver no. 17]. |
| | **Effects on Individual Time** | *"…Really I don't have enough time because I have some other things that I would like to do but because of caring for the patient I feel that the time is not enough, as you know with business when you are not there you end up losing customers and making losses…"* [Caregiver no.25]. |
| | | *"…..the time is less because I have to take care of him and when I go to the garden I don't spend there long….."* [Caregiver no.17] |
| | **Feelings after separation from their persons** | *"For sure when I am separated from her I really feel uncomfortable because one thing I know is that the parents cannot cater for her because the situation is just not all that good…………."* [Caregiver no.14] |
| | | *"…I feel bad because if I am not there, there is no one who can take care of him like I do. Though they look after him they don't care of him like myself so when I am away from home I worry a lot but I leave everything into God."* [Caregiver no.11] |
| | **Behavioral influence associated with Caregiving** | *"My behavior is influenced negatively I feel out casted, I am humble and ready for any action that may occur on him physically. I am humbled because I cannot express myself in public some people may end up talking to me badly and everyone may think that I have this disease as well……."* [Caregiver no.25] |
| | | *"…I am not always happy, I am pre-occupied with thoughts….."* [Caregiver no.27] |

*Effects of caring for a person afflicted with epilepsy on the personal relationship*. Feelings of being looked down upon or belittled for taking care of their loved one, and public fears from the belief that epilepsy is infectious, were expressed by the participants, illustrated in Table 1.

However, some participants admitted that they were being well cared for:

*"…they help me, they don't fear there is no problem."* [Caregiver no.10].

*Effect of caregiving on individual's time*. Caregivers acknowledged that their time is affected taking care of their loved ones as the patient needs more attention than any other business illustrated in Table 1.

*Feelings after separation from their persons*. Caregivers had mixed feelings while they were separated from their loved ones; experience of bad feelings when separated because of the close affection that they had developed with their person resulting from the unpredictable attacks, emerged among the participant, Table 1.

However, a sense of hopefulness had not left others in that all was well while they were away from their persons:

*"…I leave him with my daughter in-law she is good and she takes good care of him just like me……… so I do not feel very worried when I am away from him to do other activities…."* [Caregiver no.4].

Others expressed no worries at all because they had trust in God and the antiepileptic medication that they were using.

*Behavioral influence associated with caregiving*. This included positive or negative attitudes that the caregivers developed while caring for their persons, some behavioral changes getting annoyed when the patient refuses to go back for the regular clinic review, and feeling shocked especially when the attack recurs, surfaced among the participants, illustrated in Table 1.

However, a sense of good behavior persisted among the participants. This was majorly affiliated with the proper understanding of the patient's condition resulting from experience with the same patient:

*". . .Since I understand her condition, and I know that no one calls for a disease my behaviors have never changed in fact I treat her with a lot of care as a delicate person. . ."* [Caregiver no.30].

**Social burden.** This was measured based on the understanding linked to outcomes such as; effects of epilepsy on public relationships, opinions about unpredictable epileptic attacks, feelings of stigma, alternative caregivers in case of absence, and effects of caregiving to persons living with epilepsy in a family:

*Effect of caring for a person afflicted with epilepsy on public relationship*. Feelings of bad public relations emerged among the participants. This was expressed as feeling uncomfortable in public, public laughter especially when the patient is soiled for example with urine after an episode of attack, and people running away especially when the attack comes, illustrated in Table 2. However, intact public relations persisted among some participants:

*Opinions about the unpredictable epileptic attacks*. The participants were interviewed to seek their opinion about the unpredictable epileptic attacks. Concern about the situation stood out among the participants. This was majorly attributed to fear of physical harm at the height of the attack, illuminated in Table 2.

However, some expressed no feelings of worry because they were used to the situation.

*Feelings of stigma*. The participants were assessed based on the stigma that is caused while caring for their loved ones. Experiences of discrimination in society while caring for their loved ones emerged among the participants. This was linked to public fears that the condition could be infectious, the patient being looked on as "mad", illuminated in Table 2.

However, life remained normal among some participants:

*". . .Me I don't mind about people's words I just help him as a brother some people tend to isolate him but me I can't, run away from him because they say blood is thicker than water, I know the truth about this disease. . .. . ."* [Caregiver no.20].

*Alternative caregivers in case of absence*. This situation was assessed based on the caregivers' opinion concerning their absence from their persons in unavoidable circumstances.

Fear of unknowns during their absence persisted among the participants, illustrated in Table 2.

However, some caregivers expressed no worry because they have good substitute caregivers when they are away from home.

*Effects of caring for persons afflicted with patients in a family*. Sometimes families get affected while caring for their loved ones; however, some families may not be affected by the situation. Negative effects expressed as 'unsettledness' in their families as a result of caring for their loved ones emanated among the participants. This resulted from factors like stigmatization in the community, a feeling of defeat, and disparate given that almost no interventions seem to help, shown in Table 2.

**Table 2. Social burden experienced by caregivers.**

| Major Themes | Sub-themes | Selected illuminative quotes |
|---|---|---|
| Social Burden | Affected public relationship | "…That is so shameful I feel sorry because if she is in a public place and she makes noise, she begins fidgeting and eventually she falls down. Upon recovering, she gets up with a lot of energy like a mad woman and urinates on herself when this happens many people laugh at me and run away……." [Caregiver no.7]. |
| | | "…It isolates you, it segregates you they say that this one aha!…….. you look at the patient she is badly off they really categorize you to be suffering from the same condition and you don't feel like being with them. Someone can even say that you will give that thing to us. People don't want to be with you…" [Caregiver no.11]. |
| | Feelings about an unpredictable attack | "…Obviously those attacks scare me utmost, it mainly scares me when I struggle to do what I am supposed to do and after doing it you see the thing coming again, at times I also feel stressed I ask myself if by now I am around and I am here what if she is alone in the room and it so happens that it comes then what could have happened…" [Caregiver no.14]. |
| | | "I don't feel okay because I am not prepared for these attacks, I feel a lot of pain….. if there were some signs showing me that my patient is going to get an attack, I would make her sit or lie down in a safer place before it occurs so that she does not get injured…." [Caregiver no.11]. |
| | Feelings of stigma | "…It affects me sometimes people run away some people talk about him, though I don't hear them talk about him other people tell me but I trust God who brought this disease I don't have what to do I leave him to God…" [Caregiver no.14]. |
| | | "….the truth is people say that this disease is demonic but "they don't care about talks me I go ahead and take care of my person, I get for her drugs like I have come and we pray for her…" [Caregiver no.19]. |
| | Alternative caregivers in case of absence | "…When I get a problem, I first take care of her, I prepare food for her and then I go away. When I am supposed to spend a night away from home, I excuse myself and tell them that I have a patient at home, and there is no one left behind to help her, so I get back home quickly and take care of my daughter…" [Caregiver no.8]. |
| | | "….When I am going far, I have a neighbor who helps me to look after her, only that because of the nature of this condition I always ensure that I travel back home I can't spend a night away from my home…" [Caregiver no.28]. |
| | Effects on family | "…Men leave me because I can't be there with them, when we want to have some good time together that's when the child's condition gets worse; in fact, my daughter's illness has made me fail to settle in marriage. All the men I get run away from me once they realize that she is suffering from that disease…" [Caregiver no.7]. |
| | | "….We feel sad, we have tired our level best but things are not working out we feel worried…" [Caregiver no.30]. |

However, some of the caregivers expressed no effect on their families.

**Economic burden.** This was defined in terms of the cost incurred towards medical care. Caregivers are more likely to have an economic burden. This was assessed based on some of the factors such as; family economic interference like the effects of caring for a person afflicted with epilepsy on production at work, and the effects of caregiving on financial stability:

*Interference with the caregivers' source of income.* The economic burden can affect both the patient's and the caregivers' quality of life and access to medical care. These effects can be measured positively and negatively depending on the severity of the disease. A significant impact on income sources was expressed emotionally by the participants. This was majorly described as "catastrophic" as most of their earning is spent on the patient for example purchasing

**Table 3. Economic and physical burdens experienced by caregivers.**

| Major Themes | Sub-themes | Selected illuminative quotes |
|---|---|---|
| **Economic Burden** | **Interference with the caregivers' source of income** | *"…Caring for my daughter really affects me, I don't work especially when she is sick, she needs much attention and when I don't work being a tailor, I don't get money, my customers get annoyed with me when their clothes are not ready, they quarrel at me and I end up losing many of them…"* [Caregiver no.3]. |
| | | *"…There is less time to attend to business, no hope for employment opportunity due to stigma. At times when I get a job, I end up losing it because of my brother's sickness, I absentee myself from work to look after him, my income is just affected like that…"* [Caregiver no. 25]. |
| | **Effects on Production at Work** | *"….It's from farming that I get food and money when I sell my harvests. Mmmh….when you are taking care of a person like this you cannot leave him for long or even go far I dig a little and for a short while then I get back home which affects my productivity…."* [Caregiver no.4]. |
| | | *"… Sometimes the attack can come early in the morning when it comes I have to stay at home, I can't go to the garden and I can't do any other thing the whole day…"* [Caregiver no.17]. |
| | **Effects on financial stability** | *"…..yaaa….You cannot know when the attack is coming and it can come when you don't have money. You also have other problems then you have to borrow so ideally you are affected you can lack…"* [Caregiver no.20]. |
| | | *"……I can't save anything, I will have many thoughts and worries and won't have peace at all. When you have patient you may reach somewhere and say if is to live it's fine or if he dies it's God to decide…".*[Caregiver no.23]. |
| **Physical Burden** | **Feelings of uneasiness** | *"…At times you feel like getting exhausted he gives a negative response he can refuse to eat on time before getting medication, at times when he gets an attack you think may be because he didn't take his drugs…"* [Caregiver no.15]. |
| | | *"…..It's not simple, it's hectic I am somebody who has not been used to staying in one place but now my life style has changed, sitting in one place makes me more tired.."* [Caregiver no.16]. |
| | **Physical challenges** | *"….I get fatigued but for a short time so far it has happened 3–4 times when it happens fatigue must come too because she is heavy…"* [Caregiver no.22]. |
| | | *"…..Nurse, mostly I develop headaches because I have endless thoughts…..."* [Caregiver no.29]. |
| | **Night Inconveniences** | *"…..Me I sleep well except when she gets an attack there I can't sleep, I keep monitoring her the whole night……"* [caregiver no.24]. |
| | | *"…I don't sleep well, I worry about those attacks that come at night…."* [Caregiver no.19]. |
| | **Burden of feeding** | *"…You can't get appetite if you are to eat you eat very little sometimes you just go and sleep hungry….."* [Caregiver no.23]. |
| | | *"…..I don't eat, I be thee seated without energy I even don't work and I tell God to help me…."* [Caregiver 13]. |

medications that are out of stock, and transport for the regular clinic reviews, illuminated in Table 3.

*Effects of caring for persons afflicted with epilepsy on production at work.* Many caregivers find it a rewarding task to look after their beloved ones, however, for some people being a caregiver has got a negative impact on their daily production at work. Different degrees of worry about productivity at work surfaced among the participants. This was described as having less time to concentrate at work. The peasants reported skipping some days without digging due to the attacks and the business caregivers admitted to having lost several customers due to the condition, illustrated in Table 3.

*Effects of caring for a person afflicted with epilepsy on financial stability.* Depending on the severity of epilepsy, there may be other health needs that may be so costly. Financial instability was expressed by the participants in the most extreme form which is fears of potential resultant poverty resulting from the catastrophic expenditures, and much time for work being spent caring for the patient, illuminated in Table 3.

**Physical burden.** Physical burden constitutes the area of; physical challenges that may affect the health of caregivers, inconveniences associated with caring for persons afflicted with epilepsy at night, and inconveniences of feeding one's self while caring for a person afflicted with epilepsy:

*The physical burden associated with caring for persons afflicted with epilepsy.* According to the participants, some reported that it was difficult to care for their persons afflicted with epilepsy. Feelings of uneasiness during the care of the patients were expressed by the participants. This was described as feeling physically exhausted, illustrated in Table 3.

**Table 4. Views of the key informants on the lived experiences of caregivers.**

| Themes | Selected illuminative quotes |
|---|---|
| **Perception of the way caregivers felt while taking care of their persons** | *". . .. The way I look at caregivers they feel burden because they are dealing with a chronic disease and the patient cannot do much for themselves. It is not easy for the care takers. . .."* [Key informant no.8]. |
| | *". . .For them what they feel is that they have to be there for the client to ensure that they get the right drugs, at the right time, in the right dose, they also give a report about the patients progress to the health workers. . . . . ."* [Key informant no.7]. |
| **Institutional programs that dealt with the stress-related effects on the caregivers** | *". . .We have outreach programs whereby we go out to the people with epilepsy in the community and talk to them about how they should take their medication, we tell them that epilepsy is a chronic illness and we encourage the caregivers to support their persons, we also have radio programs whereby we go on air and talk about epilepsy and clear people's misconceptions about this disease . . .."* [Key informant no.5]. |
| **Role of Health workers in caring for Caregivers** | *". . .. My role is to help these caregivers to appreciate that epilepsy is a life long illness and it is just like any other disease so taking care of a patient with this illness is just like taking care of a patient with any other illness. They have to appreciate that these fits can be controlled though it takes long and someone has to be on drugs throughout their life. . .."* [Key informant no. 6]. |
| | *". . . . .I give them talk therapy this instills positive thinking in the care givers and also train others who have patients that are newly diagnosed so that they can cope with ease. . .."* [Key informant no.9]. |
| **Resources available in taking care of epileptic patients and their caregivers** | *". . . . .We have the personnel, who I identify the patients, they treat them, and follow them up in the communities. . .."* [Key informant no.9]. |
| | *". . . . . .We have specialized personnel; drugs are available and the mental health unit is fully functional it provides services to the patients 24 hours in a day. . . . . ."* [Key informant no.3]. |
| **Effects of Caring for persons afflicted with Epilepsy on Family/social relationships** | *". . .. Financially they are strained because they have to transport themselves from the hospital, they have to buy drugs for the patient and cater for the entire family as well. . .."* [Key informant no.10]. |
| | *". . . . .Resources are wasted, money is wasted in caring for them, time is wasted in caring for them and they are stigmatized and think that the disease is contagious. . ."* [Key informant no.3]. |
| **Symptoms that caregivers get as a result of caring for their loved ones** | *". . . . . . . .. They can develop depression and they are isolated in the communities. . .."* [Key informant no.3]. |
| | *". . .. I think they can develop headache, because of thinking so much about the patient's condition that has no cure, also they can have disrupted sleep patterns because epilepsy attacks come even at night and interfere with their sleep. . ."* [Key respondent no.10]. |
| **Effects of caregiving role on Caregivers' well being** | *". . . . .. It's costly each time you have to travel to the hospital, you have to buy drugs, feed the patients and handle your own issues, poverty can set in. . .."* [Key informant no.8]. |
| | *". . .Most of them end up becoming poor because they are confused, people deceive them that traditional healers can solve their problem so they spend a lot of money and by the time they come to the hospital the patient's condition may have deteriorated. . . . . .."* [Key informant no.1]. |

*Physical challenges that may affect the health of caregivers for persons afflicted with epilepsy care.* The physical burden faced by the caregivers was based on the respondent's opinion, some of the participants had challenges associated with giving care to their patients because they had to carry their patients to a safer place after a fit, they had to monitor their movement, and some of them had difficulties with feeding, especially after their persons fell, illustrated in Table 3.

However, others reported that they were comfortable with the situation.

*Inconveniences caused by caring for persons afflicted with epilepsy at night.* Night inconveniences described as 'disrupted sleep' and 'no sleep' evolved among the participants. This was attributed to being preoccupied with thought and interrupted sleep especially when an attack recurs at night, illustrated in Table 3.

*The burden of feeding while caring for a person afflicted with epilepsy.* The burden here was not so often, this depended on the frequency of the attack and the severity of the disease. When the attack was frequent, the caregivers' feeding habits got affected and vice versa. This was attributed to loss of appetite resulting from worries, and failure to prepare meals resulting from being preoccupied with caring for the patient especially when the attack comes.

However, others never lost their appetite for food during the attack because they were used to the situation.

## Key informant health workers

**Findings.** About 10 key informants were interviewed, 5(50%) of the participants were females and 5(50%) of the participants were male, 7(70%) of them were married whereas 3 (30%) of the participants were single. With regards to education 7(70%) of the participants had attained Diploma-level education, 1(10%) of the participants was a degree holder, 1(10%) was a certificate holder and 1(10%) of the participants had studied up to senior four. The majority of the participants were of different tribes 6(60%), whereas 2(20%) of the participants were Itesot and about 2(20%) of the participants were Bagishu. The majority of the participants were Psychiatric nurses 6(60%), about 2(20%) of the participants were psychiatric clinical officers, 1(10%) of the participants was a counselor and 1(10%) of the participants was a peer support worker. Seven themes emerged from the interviews that were carried out with the key informants. The themes have been summarized in Table 4.

**Perception of the way caregivers felt while taking care of their persons.** Some of the participants thought that the caregivers felt bewitched, cursed, and unfortunate. Some thought that the caregivers developed bad feelings and they ended up giving up on the caregiving role and others thought that the caregivers felt that health workers were knowledgeable and could therefore give them good advice and treatment. They also thought that the caregivers felt that it was their obligation to care for their persons, illuminated in Table 4.

**Institutional programs that dealt with the stress-related effects on the caregivers.** Some of the participants acknowledged that the institution employed psychologists, psychiatric clinical officers, counselors, social workers, and psychiatric nurses among others who gave continuous health education. They counseled and encouraged caregivers not to give up on their role, some admitted that they had community outreach programs and radio talk shows where they sensitized the public about epilepsy, and one participant reported that they linked up the needy patient who come to the facility without caregivers to the social workers for support, illustrated in Table 4.

**Role of health workers in caring for caregivers.** Some of the participants reported that their role is to counsel patients, give health education talks, encourage drug compliance,

protect the patients and their caregivers, and encourage the caregivers to continuously take care of their persons, as shown in Table 4.

**Resources available in taking care of patients with epilepsy and their caregivers.** Some of the participants reported that antiepileptic drugs were available, some said that the institution employed, and trained health workers and others said that they received support from some non-government organizations for example JENGA (A Christian-based Non-Governmental Organization), they had good security, and beds for the patients' use, illustrated in Table 4.

**Effects of caring for persons afflicted with epilepsy on family/social relationships.** Some of the participants reported that they thought that families of the caregivers were stigmatized in the communities, and others thought that the caregivers wasted resources, and lost their jobs and social network as well, illuminated in Table 4.

**Symptoms that caregivers get as a result of caring for their loved ones.** Some of the key informants reported that they thought that caregivers got fatigued and the others thought that they presented with symptoms of stress, such as headache, worry, chest pain, backache, and sleep disruption, showed in Table 4.

**Effects of caregiving role on caregivers' well being.** Some of the participants reported that they thought that the caregivers would end up becoming poor, and some reported that they thought that the caregivers experience social, physical, and psychological burdens which manifested in the form of productivity loss at work, loss of social network, loss of appetite for food, and disrupted sleep among others, illustrated in Table 4.

## Discussions

This study aimed to determine the lived experiences of caregivers of persons with epilepsy attending the epilepsy clinic at Mbale regional referral hospital, in eastern Uganda. This study discusses the lived experiences of caregivers into four interrelated themes. These include psychological, social, physical, and economic burdens. The caregivers perceived epilepsy as a major burden interfering with different aspects of their lives.

Different psychological burdens surfaced among caregivers. Worries about day-to-day living and future life of their persons were expressed by the participants. This finding has been reported in a scoping review where it was found that pervasive fear associated with the unexpected and uncertain nature of parenting a child with epilepsy exited among the caregivers [32]. The possible explanation is that managing the day-to-day care of a person with epilepsy can be exacerbated by the often unpredictable nature of the condition [33]. The other factor that emerged among the participants is a disruption in sleep and no sleep also. What could explain this sleep interference is that studies have reported the reason for the sleep disruption among caregivers to be fear of sudden unexpected death in epilepsy during sleep, and helping the patient with toileting among other needs [34]. The finding on psychological burdens reported in this present study has been reported in other related studies: a study in the United States [18], found that caregiving to patients with epilepsy adversely affects the caregiver's psychological health with many caregivers reporting anxiety, depression, and insomnia after becoming caregivers. Studies in Nigeria [35] similarly found high levels of emotional distress among caregivers of patients with epilepsy. Psychological burdens among caregivers of persons with epilepsy have been reported in other related studies [9, 36, 37]. The possible explanation for this psychological distress could be the unpredictability of seizures, fear of stigma, and unawareness of epilepsy [38].

Several social burdens in the forms of stigma and discrimination were experienced by the caregivers; disrupted public relationships, feelings of shame, and being under looked or

despised because of caring for persons afflicted with epilepsy emerged. Related studies in the United States [15] reported stigma among caregivers of patients with intractable epilepsy. These study findings are in line with the findings of other related studies elsewhere [36, 38–40] where different social consequences of epilepsy on the caregivers were reported. The reason for these disruptions in social relationships could be attributed to the beliefs, myths, and misconceptions of the public towards epilepsy. Evidence from the Ugandan community showed that the public still had beliefs that epilepsy was contagious and spiritual and that it needs traditional healers or prayers [41], More evidence from Uganda from rural and urban areas demonstrated limited knowledge in regards to epilepsy among the general public [42]. A systematic review in Sub-Saharan Africa further demonstrated that there is a substantial misconception, negative attitude, and stigma surrounding epilepsy in this region where Uganda falls [43].

This study showed that caregivers of persons afflicted with epilepsy had more economic burdens since they had to spend most of their time caring for their persons. This impacted negatively their daily functioning. This finding is in line with that of other related studies [1, 44] which showed that epilepsy was noted to have significant economic implications in terms of healthcare needs, premature death, and lost work productivity. This study finding further agreed with the findings from a systematic review in Sub-Saharan Africa [45] which showed that caregivers gave an account of spending money to care, losing some days of work for care-giving, and traveling to seek care for relatives. Absenteeism was associated with caregivers leading to the loss of jobs in some instances. High costs resulting from the care of persons with epilepsy which occur either directly or indirectly have been reported in other related studies [18, 45, 46]. More evidence has demonstrated that the social determinants of health which include financial status, education, and employment can impact a family and caregiver's well-being at the time of diagnosis of epilepsy [47]. However, the non-contingency of epilepsy further impacts these determinants with loss of job, income, and opportunities for higher education especially for the youthful patients [47]. This study found that caregivers experience various physical burdens; fatigue, chest pain, and headache. This finding agrees with the findings of a related study [18] which showed that family caregiver symptoms such as fatigue, headaches, and joint and muscle pains. The possible explanation for this is that persons with epilepsy especially children often have comorbidities like cerebral palsy and intellectual developmental delay and therefore need supervision and assistance in activities of daily living like feeding, bathing, taking medicines, communication, and mobility, thus increasing physical and emotional dependence on parents which results into a high level of parental stress [48].

The majority of the key informants reported that they administer treatment to the patients, offer counseling services to patients, and they give continuous health education talks at the facility, and conduct radio talk shows. However, findings from the central part of Uganda among the epilepsy patients at Mulago national referral hospital found limited patient-health-worker interaction [49]. They admitted having enough stocks of antiepileptic drugs and an adequate staff, they also reported that some of the patients and their caregivers were receiving support from the social workers and some Non-Governmental Organizations. The majority of the key informants believed that caregivers are stigmatized in the communities, some of them reported that the caregivers wasted resources (money and time) while caring for their persons and had lost their jobs and social network. They also thought that caregivers got symptoms like fatigue, stress, headache, and sleep disruptions these findings matched with the findings of most of the studies done in other parts of the world [39, 40].

## Study strengths and limitations

Employing in-depth interviews with the caregivers and key informants facilitated the achievement of a depth understanding of the lived experiences of the caregivers of persons with epilepsy in this social context.

Findings from this study on the lived experience of caregivers of persons with epilepsy have to be interpreted with some limitations put into consideration. The first is the influence of the researchers' personal biases. This involved researchers' personal experiences and prejudices that could have had some influence on the understanding of the lived experience of the caregivers of persons with epilepsy. To counteract this bias, open and honest discussions were held among the investigators to help elicit these preconceived ideas before data collection. Secondly, 'bracketing' [50] was done. This is where the researchers put aside preconceptions and personal knowledge when listening to the caregivers and during a reflection on the lived experiences of the caregivers. Furthermore, neutral nonverbal behavior was practiced during the interview process. Consequently, employing these strategies allowed the essence of the lived experiences to emerge naturally from the perspective of the caregivers.

Secondly, this study was conducted in a single regional referral hospital, and therefore caregivers of persons with epilepsy in other settings of Uganda could be having different experiences. For example, those who seek care from the national mental and psychiatric referral hospital and those who seek care from private facilities. Therefore, the study being single-centered and the small purposive sample recruited hampers the transferability of the study findings to other settings. However, these limitations are counteracted to a certain degree by the application of rigorous and robust qualitative methods described in the study and stick adherence to evidence-based standardized criteria for reporting qualitative research particularly the Consolidated Criteria for Reporting Qualitative Research (COREQ) [28]. Furthermore, evidence has demonstrated that purposive sampling is the most appropriate method for the selection of participants for studies that aim to understand and describe a particular phenomenon from the perspective of those who have experienced it [51]. Concerning sample size, a size of 10 to 20 is adequate provided the participants can provide a rich description of the phenomenon [52]. Therefore, our sample of 30 caregivers and 10 key informants was adequate.

## Implication for practice

Given the findings from this study that caregivers of persons with epilepsy experience physical, social, psychological, and economic burdens in attempts to deliver care to their loved ones, in clinical practice for people with epilepsy, special attention needs to be targeted at caregivers of persons with epilepsy in regards to their holistic health. The approach to be used is the bio-psychosocial-economic model [53, 54]. So, attention is paid not only to biological or physical health but also to psychological, social, and economic health through a comprehensive assessment, treatment, and counseling in the form of psychoeducation, and the formation of caregiver support groups to buffer them from feelings of burden. This will enhance the significant reduction in the burden of epilepsy experienced by the caregivers of persons with epilepsy and consequently improve their overall quality of life.

## Conclusion

This study looked at the lived experiences of caregivers of persons afflicted with epilepsy attending the epilepsy clinic at Mbale regional referral hospital.

The findings of this study have demonstrated that caregivers majorly describe epilepsy as a major burden affecting different aspects of their lives. The experiences were described in terms of psychological burdens like affected personal relationships, social burdens like stigma,

economic burdens like interference with caregivers' source of income, and finally physical burdens like night inconveniences such as sleep disruption.

## Recommendations

Policies need to be established to mitigate the psychological and social burden arising from stigma through the provision of support to overcome both experienced stigma and internalized stigma for persons with epilepsy and their caregivers. A systematic review on stigma reduction in epilepsy demonstrated to one of the important strategies to address epilepsy-related stigma is the development of policies on outreach programs to community leaders and members to shift harmful norms about epilepsy through community dialogs, as well as engagement of local leaders to share stigma messages [55].

The economic burden experienced by the caregivers can be addressed through the establishment of policies that allocate resources and implementing them for example the Uganda government development programs like the established 'parish development model' [56] should also be targeted at caregivers of persons with epilepsy to reduce the economic burden experienced by these caregivers.

Resource allocation to persons with epilepsy and their caregivers can further aid connection with the larger social environment and therefore enhance healthcare services received by these vulnerable groups of persons.

Furthermore, there is a need for policies that favor the utilization of public health models that have comprehensively tackled the HIV burden and HIV stigma in Uganda. This will provide an avenue to reduce epilepsy burden among the caregivers, and improve care and consequently quality of life of persons with epilepsy. This can be done through for example mass sensitization of the general public about epilepsy as its being done for HIV/AIDS.

Healthcare providers should continue targeting caregivers of persons afflicted with epilepsy in their service provision to ensure that their state of health is not adversely affected. Services and plans targeting patients with epilepsy need to put into consideration the burden that caregivers encounter to comprehensively manage epilepsy and its resultant burden.

We finally recommend Longitudinal studies that can study the burden experienced by caregivers of patients with epilepsy, and associated factors and offers solutions or consider the determinants of caregiver burden and its impact on quality of life (QOL) for caregivers.

## Supporting information

**S1 Data.**
(ZIP)

## Acknowledgments

We are grateful to the study participants.

## Author Contributions

**Conceptualization:** Lindah Okiah, Samuel Olowo, Stanely J. Iramiot, Rebecca Nekaka, Lydia V. N. Ssenyonga.

**Data curation:** Lindah Okiah, Samuel Olowo, Stanely J. Iramiot, Rebecca Nekaka.

**Formal analysis:** Lindah Okiah, Samuel Olowo, Stanely J. Iramiot, Rebecca Nekaka, Lydia V. N. Ssenyonga.

**Investigation:** Lindah Okiah, Samuel Olowo, Stanely J. Iramiot, Rebecca Nekaka, Lydia V. N. Ssenyonga.

**Methodology:** Lindah Okiah, Samuel Olowo, Stanely J. Iramiot, Rebecca Nekaka, Lydia V. N. Ssenyonga.

**Resources:** Lindah Okiah, Samuel Olowo, Stanely J. Iramiot, Rebecca Nekaka, Lydia V. N. Ssenyonga.

**Supervision:** Stanely J. Iramiot, Rebecca Nekaka, Lydia V. N. Ssenyonga.

**Validation:** Lindah Okiah, Samuel Olowo, Stanely J. Iramiot, Rebecca Nekaka, Lydia V. N. Ssenyonga.

**Visualization:** Lindah Okiah, Samuel Olowo, Stanely J. Iramiot, Rebecca Nekaka, Lydia V. N. Ssenyonga.

**Writing – original draft:** Lindah Okiah, Samuel Olowo, Stanely J. Iramiot, Rebecca Nekaka, Lydia V. N. Ssenyonga.

**Writing – review & editing:** Lindah Okiah, Samuel Olowo, Stanely J. Iramiot, Rebecca Nekaka, Lydia V. N. Ssenyonga.

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
