## [Decision Letter · Decision Letter 0]

5 Dec 2022

PONE-D-22-23983Lived experiences of caregivers of persons with epilepsy attending an epilepsy clinic at a tertiary hospital, eastern Uganda: A phenomenological approachPLOS ONE

Dear Dr. OLOWO,

Thank you for submitting your manuscript to PLOS ONE. After careful consideration, we feel that it has merit but does not fully meet PLOS ONE’s publication criteria as it currently stands. Therefore, we invite you to submit a revised version of the manuscript that addresses the points raised during the review process.

Specifically, the reviewers raised substantial concerns. Please see the comments in details below and provide your point-by-point responses.

We look forward to receiving your revised manuscript.

Kind regards,

Jianhong Zhou

Staff Editor

PLOS ONE

Journal Requirements:

No funding was received towards this study however the researchers do acknowledge support Busitema University Directorate of Graduate Studies, Research and Innovation (DGSRI) for small grants programs that facilitated the writing of this manuscript.  

Reviewers' comments:

Reviewer's Responses to Questions

**Comments to the Author**

1. Is the manuscript technically sound, and do the data support the conclusions?

Reviewer #1: Yes

Reviewer #2: Yes

Reviewer #3: Yes

2. Has the statistical analysis been performed appropriately and rigorously? 

Reviewer #1: Yes

Reviewer #2: N/A

Reviewer #3: Yes

3. Have the authors made all data underlying the findings in their manuscript fully available?

Reviewer #1: Yes

Reviewer #2: No

Reviewer #3: No

4. Is the manuscript presented in an intelligible fashion and written in standard English?

Reviewer #1: Yes

Reviewer #2: Yes

Reviewer #3: No

5. Review Comments to the Author

Reviewer #1: The authors set out to study the lived experiences of caregivers for PWE and address an important topic. However, some areas need improvement to improve understanding.

1. Abstract: needs to be focused and highlight some of the key results got from the study instead of just summarizing in one sentence the findings.

2. Introduction needs refocusing and emphasize the caregiver burden. Some of the text is not required or needs to be summarized.

4. Add a reference for lines 77-78.

4. Eligibility criteria indicates accessible health workers yet the study objective doesn’t include this. Please clarify.

5. Results. Remove the zeros write 9 instead of 09 etc throughout the text.

6. Some of the quotes are too long, they need to be shortened like lines 213-217.

7. The results indicate results from key informants, who are they and how were they selected.

Discussions are too long, please refocus them on caregivers burden.

Reviewer #2: Thank you for the opportunity to review this manuscript focused on understudied lived experiences of caregivers of patients with epilepsy in Uganda. Findings will improve care of both the patients and supports for family caregivers.

Here are suggestions for strengthening the manuscript:

Lines 67-68. review and revise sentence structure. missing a word or comma?

Line 74. stay consistent with either caregivers or care giver.

Line 83-84. the numbers are a bit confusing! where did the remaining (831-295) patients go? were they admitted to inpatient?

Line 87-88. need reference to support this! or is your conclusion about insufficient knowledge of caregiver experiences solely in this one clinic?

Lines 111-112: Introduction of health workers was a surprise. Consider introducing this earlier say in purpose statement. You could say that the caregivers lived experiences will be elicited directly from them and their health workers who work with them in care of the patients. Alternatively give an explanation or the why for their inclusion somewhere in lines 109-120.

Line 123. no reference was provided in support of sample size and saturation info.

Line 132-133. this is a surprise! give more context and detail, such as rationale etc.

Line 151. review and revise this sentence. maybe missing some words.

Line 166-167. review and correct sentence.

Line 244-247. the quote does not seem to fit the sub-theme description preceding it.

Line 325-330. this section was a bit hard to follow.

Line 362. incorrect use of ; need to review entire document for greater grammar.

Line 540. use person-centered or strength-based language, like the patient or person with epilepsy instead of epileptic.

Line 551. Correct heads to headache.

Reviewer #3: Review Comments

This paper highlights the lived experiences of care givers of persons with epilepsy attending the epilepsy clinic at Mbale regional referral hospital of eastern Uganda. I am glad to have reviewed it.

Overall: This is an engaging paper with a well written abstract, a clear aim, well described methodology and well-presented findings that suggest that caregivers of persons with epilepsy various burdens, including psychological, social, economic, and physical burdens.

Abstract and Introduction: The authors have provided a well written abstract, summarising the research question and the key findings. The authors took time to identify existing literature on the topic and justified how their study relates to the available literature. However, these sections could benefit from proofing of the original word document to address any grammatical and typographical errors.

Methodology: The methods are described in sufficient details, making the study technically sound although I am not an expert on the method of data analysis used for this particular study. This section could benefit from proofing of the original word document for rephrasing of some of the sentences (for example, sentence 166-168) to address grammatical and typographical errors.

Results, Discussions and Conclusion: The results are well presented in the various thematic areas that address the aim of the study and support the conclusion of the study. However, the study fails to discuss the limitations of the approach used to address the aim of the study and the implications of the findings of the study in the discussion. Thus, the authors should revisit the discussion to acknowledge the possible strengths and limitations of the study (Researcher or Interviewer induced bias, Validity Vs Reliability, Language Barrier, highly qualitative nature of findings), highlight how the limitations may have been addressed and include the implications for practice of the findings of the present study to Uganda and Global Community. In addition, the authors should consider proofing all the sections of the paper to address grammatical and typographical errors to improve the flow and readability of the text. For instance, sentence 569-579 appears to be incomplete and this means the conclusion may require rephrasing.

Recommendations: While the recommendation for continuity of current activities being done by the mental health practitioners for patients with epilepsy and their care givers is a good idea, it would be good to make recommendations of solutions that address policy initiatives around the different themes of burdens experienced by the care givers and make recommendations for future research that may provide additional data on the burden being experienced by caregivers for patients with epilepsy. For instance, the authors could consider making suggestions of Longitudinal studies that can study the burden experienced by caregivers of patients with epilepsy, associated factors and offers solutions or consider the determinants of caregiver burden and its impact on quality of life (QOL) for caregivers.

Thank you for providing this piece of research. I am certain that with the above suggested revisions, it will be a great paper and worth the read.

6. PLOS authors have the option to publish the peer review history of their article (what does this mean?). If published, this will include your full peer review and any attached files.

Reviewer #1: No

Reviewer #2: No

Reviewer #3: No

---

## [Author Response · Author response to Decision Letter 0]

15 Jan 2023

The responses to the specific reviewer and editor comments are in the "response to reviewers" file uploaded together with the revised manuscript file.

---

## [Editor Report · Decision Letter 1]

24 Jan 2023

PONE-D-22-23983R1Lived experiences of caregivers of persons with epilepsy attending an epilepsy clinic at a tertiary hospital, eastern Uganda: A phenomenological approachPLOS ONE

Dear Dr. OLOWO Samuel,

Thank you for submitting your manuscript to PLOS ONE. After careful consideration, we feel that it has merit but does not fully meet PLOS ONE’s publication criteria as it currently stands. Therefore, we invite you to submit a revised version of the manuscript that addresses the points raised during the review process.

We look forward to receiving your revised manuscript.

Kind regards,

Mark Kaddumukasa

Guest Editor

PLOS ONE

Journal Requirements:

Additional Editor Comments:

The authors have responded to the comments raised by the reviewers however, some changes are needed and clarity to reduce some of the sections that are too wordy.

1. The ethical considerations section is too wordy with lots of repetitions, lines 192-194 similar with 199-201; 199-203 with 204-206. Please avoid repetitions.

2. Some of the quotes remain too long and may reduce readership, it would be better to summarize them in a table under the various themes so as to enhance easy reading.

3. Th discussion and limitations needs to be shortened and some sections deleted like lines 529-531, it would be nice to focus on the major study findings. Focus the limitations and make them concise and clear.

---

## [Author Response · Author response to Decision Letter 1]

11 Mar 2023

We have summarized the comments as highlighted by reviewers and editors and their corresponding responses from the authors in a table format accompanying the rebuttal letter.

---

## [Decision Letter · Decision Letter 2]

19 Apr 2023

PONE-D-22-23983R2Lived experiences of caregivers of persons with epilepsy attending an epilepsy clinic at a tertiary hospital, eastern Uganda: A phenomenological approachPLOS ONE

Dear Dr. OLOWO,

Thank you for submitting your manuscript to PLOS ONE. After careful consideration, we feel that it has merit but does not fully meet PLOS ONE’s publication criteria as it currently stands. Therefore, we invite you to submit a revised version of the manuscript that addresses the points raised during the review process.

Please submit your revised manuscript by Jun 03 2023 11:59PM. If you will need more time than this to complete your revisions, please reply to this message or contact the journal office at plosone@plos.org. Please include the following items when submitting your revised manuscript:A rebuttal letter that responds to each point raised by the academic editor and reviewer(s). You should upload this letter as a separate file labeled 'Response to Reviewers'.A marked-up copy of your manuscript that highlights changes made to the original version. You should upload this as a separate file labeled 'Revised Manuscript with Track Changes'.An unmarked version of your revised paper without tracked changes. You should upload this as a separate file labeled 'Manuscript'.

We look forward to receiving your revised manuscript.

Kind regards,

Mark Kaddumukasa

Guest Editor

PLOS ONE

Additional Editor Comments:

Reviewer 1

The authors need to back up their statements in the introduction, for example “the outpatient clinic recruits 295 patients in July” please reference

Eligibility criteria needs to be written well for both care takers and health care providers, how were health workers selected? Was stratification done for the participants

The authors need to discuss their results succinctly, the major findings first and then a fitting explanation, the conclusion and recommendations have to be re written, conclusions drawn from the findings and the implications too

Reviewer 2

This is an important paper that merits publication—the data/content gathered is important and valuable. However, the paper needs notable reworking, and would strongly benefit from all of the authors reviewing it with an eye towards grammar and typical publication standards (some examples below). I would suggest that a senior member of the team review and guide the rework, and that additional support be added to the team if needed.

Open ended questions with follow-up probes are typically considered a structured interveiw guide-- depends on order and flexibility with the questions.

Examples related to grammar and problems with referencing:

-in the abstract "Feelings" is midsentence and capitalized.

-The introduction states that the ILAE recognizes epilepsy as an NCD, which is true. Epilepsy is recognized as a chronic major NCD by ILAE, the World Health Organization, etc. But the reference attached to this (3) is not an ILAE publication. It would be best to reference a publication of th ILAE (or better, the WHO), and then reference the Singh paper for the worldwide statistic (best done by having these 2-3 references together at the end of the sentence).

-The incidence of epilepsy has been reported to be high in Uganda with over 156 new cases of epilepsy per 100,000 people each year and an increasing prevalence of 13.3% [7]-- this paper reports the crude prevalence ast 13.3, what paper holds the 156/100,000 rate? Please reference that primary source.

-Line 80 has 13 referenced twice

-While epilepsy is a common phenomenal in Uganda: phenomenon

-Lines 88-94 are given as justification for the study. The number of people attending the clinic, their gender, etc., should not drive a research question. Instead, lack of knowledge of caregiver's lived experience (regardless of number) drives the question.

-bed occupancy averagely 359

-147: the interviews guide was

-150: like leading question

183: "till" is not standard English

186: Conformability of the study is demonstrated by how the empiricalthe empirical data is in line with the study findings.

The Ethical considerations section is far too long. Please review recent publications for examples of typical reporting.

VERY IMPORTANT: Tables 1, 2, and 3, 4 have theme/subthemes listed and selected illumative quotes. Then the text has more quotes. This presents the data in a split fashion, when it is necessary to pick one. The tables are efficient, and thus the text should be limited to introducing the content of the table, not providing more content.

The Discussion needs to not only relate the current findings to existing papers, as it it does, but also needs to professionally and succinctly offer what NEW information was learned from this study.

Can the sample be described by patient’s age, type/duration/severity of epilepsy? These factors could influence caregiver burden and should be reported.

Recommendations should go in the Discussion unless this journal routinely breaks it into a separate section.

“The collection and analysis of the data were done simultaneously”. I do not think this is accurate. Thematic analysis requires transcription and careful study of results, which would occur well after the discussion that collected the data.

“_an inductive thematic analysis....thorough reading and understanding of the transcrips ensured to obtain an overall sense...Line coding...A thorough review of the transcripts were ensured and later a list of codes....” This could be made more succinct and stated once. If in the methods the authors describe the thematic analysis approach well, it does not need to be stated that the content was read and understood, etc. This is where a review of the paper by the senior member of the team may be helpful to make reporting concise and clear.

Reviewers' comments:

Reviewer's Responses to Questions

**Comments to the Author**

1. If the authors have adequately addressed your comments raised in a previous round of review and you feel that this manuscript is now acceptable for publication, you may indicate that here to bypass the “Comments to the Author” section, enter your conflict of interest statement in the “Confidential to Editor” section, and submit your "Accept" recommendation.

Reviewer #4: All comments have been addressed

Reviewer #5: (No Response)

2. Is the manuscript technically sound, and do the data support the conclusions?

Reviewer #4: Yes

Reviewer #5: Yes

3. Has the statistical analysis been performed appropriately and rigorously? 

Reviewer #4: Yes

Reviewer #5: N/A

4. Have the authors made all data underlying the findings in their manuscript fully available?

Reviewer #4: Yes

Reviewer #5: Yes

5. Is the manuscript presented in an intelligible fashion and written in standard English?

Reviewer #4: Yes

Reviewer #5: No

6. Review Comments to the Author

Reviewer #4: The authors need to back up their statements in the introduction, for example “the outpatient clinic recruits 295 patients in July” please reference

Eligibility criteria needs to be written well for both care takers and health care providers, how were health workers selected? Was stratification done for the participants

The authors need to discuss their results succinctly, the major findings first and then a fitting explanation, the conclusion and recommendations have to be re written, conclusions drawn from the findings and the implications too

Reviewer #5: This is an important paper that merits publication—the data/content gathered is important and valuable. However, the paper needs notable reworking, and would strongly benefit from all of the authors reviewing it with an eye towards grammar and typical publication standards (some examples below). I would suggest that a senior member of the team review and guide the rework, and that additional support be added to the team if needed.

Open ended questions with follow-up probes are typically considered a structured interveiw guide-- depends on order and flexibility with the questions.

Examples related to grammar and problems with referencing:

-in the abstract "Feelings" is midsentence and capitalized.

-The introduction states that the ILAE recognizes epilepsy as an NCD, which is true. Epilepsy is recognized as a chronic major NCD by ILAE, the World Health Organization, etc. But the reference attached to this (3) is not an ILAE publication. It would be best to reference a publication of th ILAE (or better, the WHO), and then reference the Singh paper for the worldwide statistic (best done by having these 2-3 references together at the end of the sentence).

-The incidence of epilepsy has been reported to be high in Uganda with over 156 new cases of epilepsy per 100,000 people each year and an increasing prevalence of 13.3% [7]-- this paper reports the crude prevalence ast 13.3, what paper holds the 156/100,000 rate? Please reference that primary source.

-Line 80 has 13 referenced twice

-While epilepsy is a common phenomenal in Uganda: phenomenon

-Lines 88-94 are given as justification for the study. The number of people attending the clinic, their gender, etc., should not drive a research question. Instead, lack of knowledge of caregiver's lived experience (regardless of number) drives the question.

-bed occupancy averagely 359

-147: the interviews guide was

-150: like leading question

183: "till" is not standard English

186: Conformability of the study is demonstrated by how the empiricalthe empirical data is in line with the study findings.

The Ethical considerations section is far too long. Please review recent publications for examples of typical reporting.

VERY IMPORTANT: Tables 1, 2, and 3, 4 have theme/subthemes listed and selected illumative quotes. Then the text has more quotes. This presents the data in a split fashion, when it is necessary to pick one. The tables are efficient, and thus the text should be limited to introducing the content of the table, not providing more content.

The Discussion needs to not only relate the current findings to existing papers, as it it does, but also needs to professionally and succinctly offer what NEW information was learned from this study.

Can the sample be described by patient’s age, type/duration/severity of epilepsy? These factors could influence caregiver burden and should be reported.

Recommendations should go in the Discussion unless this journal routinely breaks it into a separate section.

“The collection and analysis of the data were done simultaneously”. I do not think this is accurate. Thematic analysis requires transcription and careful study of results, which would occur well after the discussion that collected the data.

“_an inductive thematic analysis....thorough reading and understanding of the transcrips ensured to obtain an overall sense...Line coding...A thorough review of the transcripts were ensured and later a list of codes....” This could be made more succinct and stated once. If in the methods the authors describe the thematic analysis approach well, it does not need to be stated that the content was read and understood, etc. This is where a review of the paper by the senior member of the team may be helpful to make reporting concise and clear.

7. PLOS authors have the option to publish the peer review history of their article (what does this mean?). If published, this will include your full peer review and any attached files.

Reviewer #4: No

Reviewer #5: No

---

## [Author Response · Author response to Decision Letter 2]

3 Jun 2023

The response to the reviewers have been summarized in the response to reviewer document uploaded

---

## [Decision Letter · Decision Letter 3]

21 Jun 2023

Lived experiences of caregivers of persons with epilepsy attending an epilepsy clinic at a tertiary hospital, eastern Uganda: A phenomenological approach

PONE-D-22-23983R3

Dear Dr. OLOWO,

We’re pleased to inform you that your manuscript has been judged scientifically suitable for publication and will be formally accepted for publication once it meets all outstanding technical requirements.

Please also rewrite the statement for eligibility age of consent, clarify on the selection of caregivers and repetitions in this section. 

Kind regards,

Mark Kaddumukasa

Guest Editor

PLOS ONE

Additional Editor Comments (optional):

Reviewers' comments:

Reviewer's Responses to Questions

**Comments to the Author**

1. If the authors have adequately addressed your comments raised in a previous round of review and you feel that this manuscript is now acceptable for publication, you may indicate that here to bypass the “Comments to the Author” section, enter your conflict of interest statement in the “Confidential to Editor” section, and submit your "Accept" recommendation.

Reviewer #4: (No Response)

Reviewer #5: All comments have been addressed

2. Is the manuscript technically sound, and do the data support the conclusions?

Reviewer #4: Yes

Reviewer #5: Yes

3. Has the statistical analysis been performed appropriately and rigorously? 

Reviewer #4: N/A

Reviewer #5: Yes

4. Have the authors made all data underlying the findings in their manuscript fully available?

Reviewer #4: Yes

Reviewer #5: Yes

5. Is the manuscript presented in an intelligible fashion and written in standard English?

Reviewer #4: Yes

Reviewer #5: Yes

6. Review Comments to the Author

Reviewer #4: see the comments attached in the documents

1.Rephrase/ rewrite the statement for eligibility age of consent.

2.How where caregivers selected, given the low educational status?

3.There is a lot of repetition in this section, that needs to be shortened to remove the repetition.

4. How was stratification done to ensure full coerciveness of the group and participation?

Reviewer #5: This manuscript is remarkably improved, and should be published. It is better organized and clear, and contributes to the literature. There are a few remaining items to clean up:

Rigor and Trustworthiness of the study: this section is not typically included, these practices are expected. You can eliminate or reduce to a sentence.

Confirmability: 'Audio recordings not destroyed which can enable others to track the progress.' This is not standard practice. Instead, recordings should be transcribed and destroyed, in order to reduce the likelihood of loss of confidentiality. Deidentified transcripts would then be shared!

349. 407 quotes remain in the text, rather than in textboxes. Would be better to move those into the boxes and allow the text to remain as descriptions.

Ref 42 is now missing. You could add 42 to the previous group (instead of ending -41 it would be -42) and add in references: 42: Sanchez et al, Epilepsy and Behavior, 2021).

WELL DONE REVISION

With these revisions the paper should be accepted

7. PLOS authors have the option to publish the peer review history of their article (what does this mean?). If published, this will include your full peer review and any attached files.

Reviewer #4: No

Reviewer #5: No

---

## [Editor Report · Acceptance letter]

10 Jul 2023

PONE-D-22-23983R3 

 Lived experiences of caregivers of persons with epilepsy attending an epilepsy clinic at a tertiary hospital, eastern Uganda: A phenomenological approach 

Dear Dr. OLOWO:

I'm pleased to inform you that your manuscript has been deemed suitable for publication in PLOS ONE. Congratulations! Your manuscript is now with our production department. 

Kind regards, 

on behalf of

Dr. Mark Kaddumukasa 

Guest Editor

PLOS ONE